# Stability of highly-twisted Skyrmions from contact topology

Yichen Hu[1*], Thomas Machon[2†],

**1** The Rudolf Peierls Centre for Theoretical Physics, University of Oxford, Oxford OX1 3PU, UK
**2** H.H. Wills Physics Laboratory, University of Bristol, Bristol BS8 1TL, UK
* yichen.hu@physics.ox.ac.uk
† t.machon@bristol.ac.uk

May 27, 2021

## Abstract

**We describe a topological protection mechanism for highly-twisted two-dimensional Skyrmions in systems with Dyloshinskii-Moriya (DM) coupling, where non-zero DM energy density (dubbed "twisting energy density") acts as a kind of band gap in real space, yielding an $\mathbb{N}$ invariant for highly twisted Skyrmions. We prove our result through the application of contact topology by extending our system along a fictitious third dimension, and further establish the structural stability of highly-twisted Skyrmions under arbitrary distortions with energies below the "gap". Our results apply for all two-dimensional systems hosting Skyrmion excitations including spin-orbit coupled systems exhibiting quantum Hall ferromagnetism.**

## 1 Introduction

Skyrmions and their variants (anti-Skyrmions, merons, etc.) are non-trivial spin textures characterized by a Skyrmion charge counting the number of times spin directions cover the sphere. This topological aspect of spin textures in real-space leads to their stability with physical consequences such as the topological Hall effect [1]. Skyrmion-like structures are ubiquitous in condensed matter physics, appearing in magnetic materials [2–7], cold atom systems [8–10], liquid crystals [11–15] and quantum Hall ferromagnetic systems [16–18]. Together with their solitonic nature, Skyrmions can be created, annihilated, and manipulated at high speed by extremely low current density [19, 20]. This makes them ideal for information storage and processing, particularly as Skyrmions are nanometer-sized for certain magnetic materials or thin film heterostructures. Skyrmion-based racetrack memory [21–24] is predicted to have higher storage density and lower energy consumption compared to all present-day solid state devices. Current research efforts [25–32] aim to generate more robust and enduring Skyrmions in room temperature for future applications in spintronic devices.

Skyrmions and their variants (see Fig. 1) can be generated by various mechanisms in magnetic systems such as dipolar interactions [33–35], frustrated exchange interactions [36] and four-spin interactions [5]. In this paper, we will focus on highly-twisted Skyrmions arising from Dyloshinskii-Moriya interactions [37, 38]. These inversion symmetry breaking interactions induced by spin-orbit coupling are commonly found in magnetic materials with non-centrosymmetric lattices. The DM interaction controls several key aspects of Skyrmions. Its strength determines the size of Skyrmions and its sign determines the chirality of Skyrmions (twisting direction of spin rotation). Here we show that surprisingly, DM interactions also constitute a crucial part of a novel topological protection mechanism for stability of highly-twisted Skyrmions. The first hint comes from observation of stable $2\pi$ twisted Skyrmions - "Skyrmioniums" (Fig.1) [7, 39–41]. Unlike usual Skyrmions, the total Skyrmion charge of these spin textures is zero. Conventional wisdom would suggest that non-topological Skyrmioniums are fragile as spins within them can be

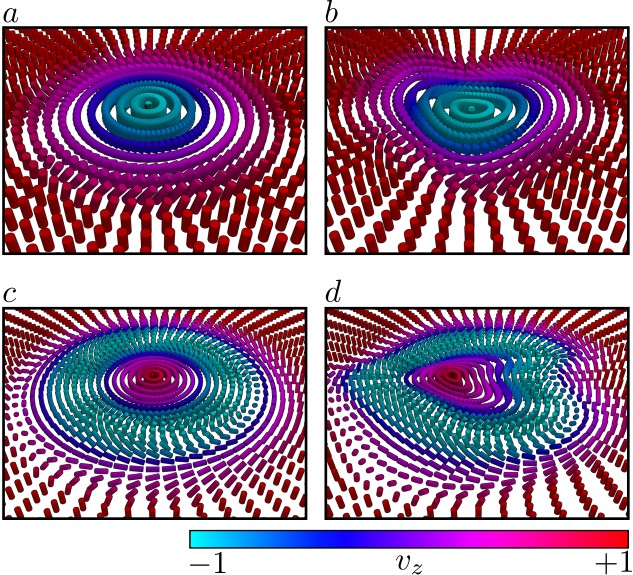

Figure 1: *Top* a: Isolated $\pi$ twisted Skyrmion with Skyrmion charge 1 in a vertical background field, b: Arbitrary twist-preserving distortion of the Skyrmion of the type considered here. *Bottom* c: $2\pi$ twisted Skyrmion with Skyrmion charge 0, d: Arbitrary twist-preserving distortion of the Skyrmion of the type considered here. Color gives the value of $v_z$.

smoothly deformed to a uniform spin configuration without any topological obstruction. However, the extra "twist" from the DM interaction completely changes this picture. Physically, the non-vanishing DM energy density ("twisting energy density") in real-space bears striking resemblance with the band gap for insulators in momentum-space [42, 43]. Same as band gap for insulators, twisting energy density is also a purely quantum phenomenon induced by relativistic spin-orbit interactions. We claim and prove in this paper that, for regions with non-vanishing twisting energy density, highly-twisted Skyrmions together with their Skyrmion charges enclosed remain invariant under arbitrary distortions with energies below this real-space "gap". These distortions are thus twist-preserving. Just like a topological insulator phase stays as a robust phase of matter provided there is a non-vanishing band gap, a highly-twisted spin texture such as a Skyrmionium stays unaltered provided there is a non-vanishing twisting energy density. Any transition to an ordinary insulator phase or trivial spin configuration must pass through at least one configuration where the band gap or twisting energy density goes to zero respectively. More intriguingly, symmetries are fundamental for each case but take opposite roles. Time-reversal and inversion symmetry [44] are essential for the existence of topological insulators and their helical edge modes while these symmetries are explicitly broken by DM interactions. By extending our 2D thin film geometry to a fictitious third direction, we turn the non-vanishing twisting energy density condition for the magnetization vector field to the contact condition for its dual 1-form. This enables us to leverage a fundamental theorem in contact topology – Gray's theorem [45, 46] – to rigorously establish the robustness of these highly-twisted Skyrmions characterized by a new $\mathbb{N}$ topological invariant.

## 2 Structural stability of highly-twisted Skyrmions

### 2.1 Energy of a Skyrmion

We start with a region $M$, which is taken as a simply connected domain in $\mathbb{R}^2$. We associate to each point in $M$ a magnetization vector $\mathbf{v}(x, y) : M \to S^2$. This is a generic model of a thin film

ferromagnet. The total energy $E$ has the following components [47]:

$$E = kE_{ke} + E_{DM} + aE_f, \tag{1}$$

$E_{ke} = \int_M |\nabla \mathbf{v}|^2$ is the kinetic energy of magnetization vector fields and $k$ is the stiffness. The second term $E_{DM} = \int_M e_{DM}$ corresponds to energy from DM interactions. For a bulk material in 3D, the local twisting energy density is simply

$$e_{DM}(x, y, z) = d\mathbf{v}(x, y, z) \cdot \nabla \times \mathbf{v}(x, y, z). \tag{2}$$

This term explicitly breaks time-reversal and inversion symmetry in real-space. The sign of $d$ and $\mathbf{v} \cdot \nabla \times \mathbf{v}$ combined together determine chirality of twisting direction for $\mathbf{v}$. As a spin-orbit coupling term, it breaks spin or orbital rotational symmetry individually, but is invariant under a simultaneous $SO(3)$ rotation of spin and space. In a thin film geometry, the full $SO(3)$ rotation in spin and orbital space gets modified to $SO(2)$ symmetry within $xy$-plane of spin and orbital space. Therefore, a generic twisting energy density term appropriate for a thin film becomes

$$
\begin{aligned}
e_{DM}(x, y) = {} & d_b \mathbf{v}(x, y) \cdot \nabla' \times \mathbf{v}(x, y) \\
& + d_n [(\mathbf{v}(x, y) \cdot \nabla') v^z(x, y) - v^z(x, y)(\nabla' \cdot \mathbf{v}(x, y))]
\end{aligned}
\tag{3}
$$

where $\nabla' = (\partial_x, \partial_y, 0)$. The first term prefers chiral (twisted) configurations such as Bloch Skyrmions whereas the second term prefers achiral Néel Skyrmions (hedgehogs). Our results apply exclusively to twisted configurations so in the following we set $d_n = 0$.

The last term $E_f$ makes sure that the minimal energy configuration of a Skyrmion has finite radius. This can be achieved by introducing shape or magnetocrystalline anisotropy or an external magnetic field which induces Zeeman energy. Using parameters of (1), the radius of a Skyrmion is $R \sim \frac{d}{a}$ [47].

## 2.2 Twist-preserving topology

In the interior of a highly-twisted Skyrmion, stabilized by the energy (1), $\mathbf{m} \cdot \nabla \times \mathbf{m}$ is non-zero everywhere. Suppose now the soliton is moved by the action of some external field or other operation. We say these distortions of the texture are *twist-preserving* provided $\mathbf{m} \cdot \nabla \times \mathbf{m}$ remains non-zero inside the interior region of the soliton, even if the overall structure is in motion.

As shown in Fig. 2, the condition $\mathbf{m} \cdot \mathbf{e}_z = 0$ defines a set of closed curves in the plane, where the magnetisation points in the $xy$ plane. Between these curves are regions, where the $z$ component of $\mathbf{m}$ is either positive or negative. The topology of this set of curves together with the sign of $m_z$ in between remain intact under arbitrary twist-preserving distortions. This is rigorously proven in Section 2.3.

With the background field pointing upwards, and positive twisting energy density (left-handed configurations), the charge $Q$ of a highly-twisted $n\pi$ Skyrmion is $n \mod 2$. This means that, in principle, the transformation $n \rightarrow n \pm 2m$ does not change the Skyrmion charge, and hence can be achieved via a non-singular transformation. However, requiring that twisting energy density remains non-zero within the Skyrmion, *i.e.* a twist-preserving distortion, the topological stability ensures that this transition cannot happen, and the number $n \in \mathbb{N}$ becomes a *bona-fide* topological invariant of the Skyrmion.

This topological structure of highly-twisted Skyrmions also gives structure to the Skyrmion charge density *within* the Skyrmion itself. Per our structural result, any Skyrmion can be considered as a set of nested bands with a central disk and outer band, defined by the set $\Gamma$. In principle more exotic arrangements may exist, which we do not consider here. Our result implies that the Skyrmion charge of each band is well-defined – our structural stability result means that Skyrmion

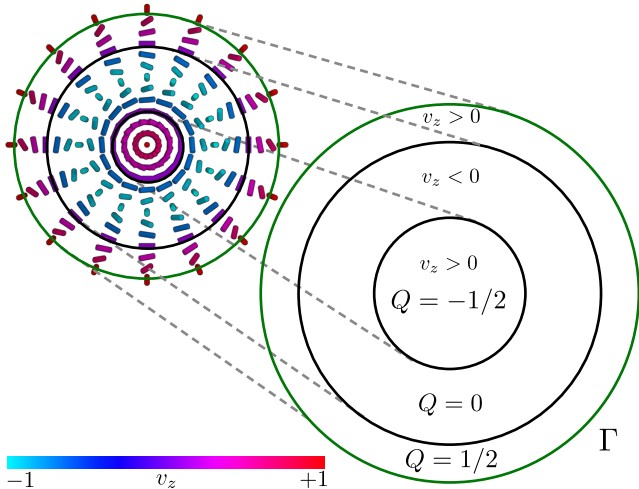

Figure 2: Anatomy of a chiral $2\pi$ Skyrmion. The set of curves $\Gamma$, comprises the places where $v_z = 0$ (black) as well as the boundary of the Skyrmion (green). The topology of the set of curves $\Gamma$ and regions in between is preserved under arbitrary twist-preserving distortions. Moreover, the Skyrmion charge within each region is also preserved.

charge cannot 'leak' between regions. Consider the spin field $\mathbf{v}$ as a map $M \to S^2$, then the Skyrmion charge in a region $R \subset M$ is the pullback of the area form $\omega$ under $\mathbf{v}$ integrated over $R$,

$$Q_R = \int_R \mathbf{v}^*\omega = \frac{1}{4\pi}\int_R \epsilon_{abc}v^a\epsilon^{ij}\partial_i v^b \partial_j v^c \, dxdy. \tag{4}$$

Alternatively, $Q_R$ is the oriented area of the image of $\mathbf{v}(R)$ on $S^2$. If the map $\mathbf{v}$ is deformed, then the change in $Q_R$ is the change in this oriented area. By the Gauss-Bonnet theorem, if the boundary of $\mathbf{v}(R) \subset S^2$ is constant in time, then $Q_R$ must also be constant. Now take $R$ to be one of the regions of the Skyrmion $S$. These regions move over time, but the area on $S^2$ swept out must remain identical, since $\mathbf{v}$ on the bounding curves $\Gamma$ does not change – on the boundary of the Skyrmion it points vertically, and on the curves defined by $v_z = 0$ it sweeps out an equatorial great circle. It follows that the Skyrmion charge within each region is constant. This is illustrated in Fig. 2.

More generally, for an $n\pi$ twisted Skyrmion there are three types of regions defined by $\Gamma$ – a central disc, an outer band, and $n-1$ inner bands. Let $\sigma \in \{-1,1\}$ be the sign of the $z$ component of the background field, and $\tau \in \{-1,1\}$ the sign of twisting energy density, then a short calculation shows that the total Skyrmion charge in the outer band is $\sigma\tau/2$, in the inner bands it is zero, and in the central disc it is $\sigma\tau(-1)^{n-1}/2$. For example in Fig. 2 we have $\sigma = +1$, $\tau = +1$, and $n = 2$, giving a total charge of zero.

## 2.3 Structural stability theorem

We now derive the DM-protected structural stability of a Skrymion-like soliton. Our argument assumes a single isolated object surrounded by a uniform background, however the construction can be easily extended to an arbitrary collection of Skyrmions, assuming they do not overlap.

The configuration is defined by a time-dependent magnetization vector $\mathbf{v}(x,y)$ at each point in $M$, which we assume varies smoothly with $x$ and $y$. We also assume that at each time $t$ there is a sub region $S_t \subset M$, representing the Skyrmion or similar twisted soliton, with boundary a smooth topological circle, such that within $S_t$, $\mathbf{v} \cdot \nabla' \times \mathbf{v} \neq 0$ at all times and outside $S_t$, $\mathbf{v} = \mathbf{e}_z$ points in the $z$-direction[1].

---

[1] There is a further technical assumption that $S_t$ never touches the boundary of $M$

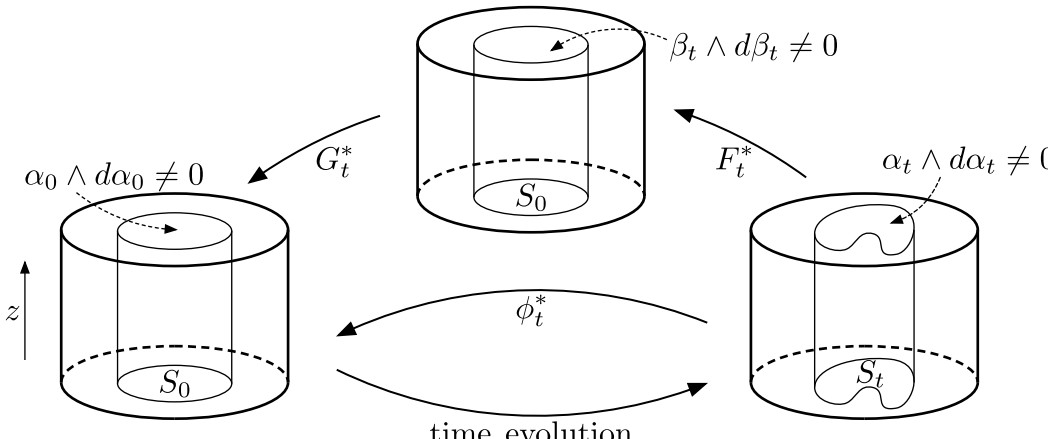

Figure 3: Construction of structural stability for chiral Skyrmions. The initial Skyrmion, or similar twisted soliton, is contained in the region $S_0$ and we artificially extend the system symmetrically in the $z$ direction (bottom left). The non-zero DM density within $S_0$ is given by $\alpha_0 \wedge d\alpha_0 \neq 0$. Under time evolution that preserves the contact condition *within* the region $S_t$, but which is otherwise arbitrary, the Skyrmion is contained within the region $S_t$ (bottom right). We then construct a diffeomorphism $\phi_t$ of the domain, so that the pullback of $\alpha_t$ under $\phi_t$ is $\alpha_0$. The pullback $F_t^*$ of $\alpha_t$, gives $\beta_t$, with the Skyrmion contained in $S_0$ (center) top. Pulling back by $G_t$ then yields $\alpha_0$.

We now artificially extend $\mathbf{v}$ along the third dimension $z$ to $M$ such that $\mathbf{v}$ at a fixed $(x, y)$ is constant along the $z$-direction. We then define a new non-singular unit vector field $\mathbf{m}(x, y, z) \equiv \mathbf{v}(x, y)$ for all $z \in \mathbb{R}$. The previous twisting energy term $\mathbf{v} \cdot \nabla' \times \mathbf{v}$ now becomes

$$\mathbf{m} \cdot \nabla \times \mathbf{m}, \tag{5}$$

on $M \times \mathbb{R}$. In three-dimensional Euclidean space, the condition of a non-vanishing twisting energy term $\mathbf{m} \cdot \nabla \times \mathbf{m} \neq 0$ can be reformulated in terms of its dual 1-form $\alpha$ as the contact condition [45]

$$\alpha \wedge d\alpha \neq 0, \tag{6}$$

where component-wise $(\alpha)_i = \sum_j g_{ij} m^j$ and $g_{ij}$ is the Euclidean metric ($i, j \in \{x, y, z\}$). Using the 1-form $\alpha$ rather than $\mathbf{m}$ is done for technical reasons – the construction below requires using the covariant form of $\mathbf{m}$. Let $\alpha_t$ denote $\alpha$ at time $t$. We now adapt Gray's theorem [45,48] to show that, under our assumptions, there is a time-dependent coordinate transformation $\phi_t : M \times \mathbb{R} \to M \times \mathbb{R}$ such that

$$\phi_t^* \alpha_t = \lambda_t \alpha_0, \tag{7}$$

where $\phi_t^* \alpha_t$ is the pullback of $\alpha_t$ under $\phi_t$ and $\lambda_t$ is a smooth positive function. This states that provided the interior of the Skyrmion has non-zero twisting energy density, the time evolution of the system is equivalent to a coordinate transformation ($\phi_t$) under which $\mathbf{v}$ transforms covariantly. The particular form of $\phi_t$ is not important, but the fact that it exists implies that various topological and geometric properties of $\mathbf{v}$ are preserved.

To establish this we first construct an arbitrary diffeomorphism $f_t : M \to M$, equal to the identity on $\partial M$, such that $f_t(S_0) = S_t$. Extending $f_t$ to a map $F_t : M \times \mathbb{R} \to M \times \mathbb{R}$ as the identity in the $z$ direction, the 1-form $\beta_t = F_t^* \alpha_t$ restricted to the interior of $S_0$ then defines a contact structure in $S_0 \times \mathbb{R}$, with $\beta_0 = \alpha_0$ and $\beta_t = a_t dz$ on the boundary of $S_0$, for some positive constant $a_t$. By Gray's theorem, there is then a diffeomorphism $G_t : S_0 \times \mathbb{R} \to S_0 \times \mathbb{R}$, restricting to the identity on $\partial S_0 \times \mathbb{R}$, such that $G_t^* \beta_t = \mu_t \beta_0$, for some positive function $\mu_t$. We

can extend the map $G_t$ to the entirety of $M \times \mathbb{R}$ by making it the identity outside of $S_0 \times \mathbb{R}$. The diffeormorphism $\phi_t = F_t \circ G_t$ then satisfies $\phi_t^* \alpha_t = G_t^* F_t^* \alpha_t = \lambda_t \alpha_0$, establishing (7).

This tells us that deformations of a Skyrmion, or similar twisted soliton, that preserve a non-zero twisting energy density are equivalent to diffeomorphisms. We now exploit the fact that our diffeormorphisms have a $z$-symmetry, to derive further structural properties. Explictly writing the diffeormorphism $\phi_t$ in terms of primed coordinates gives the following functional dependencies for $\phi_t^{-1}$

$$x(x', y'), \ y(x', y'), \ z(x', y', z'). \tag{8}$$

Both the $x$ and $y$ directions transform without dependence on $z$. In components, (7) tells us that $\alpha_t$ transforms as

$$(\alpha_t)_i = \frac{\partial x^j}{\partial (x')^i} (\alpha_0)_j. \tag{9}$$

But since neither $\alpha_0$ nor $\alpha_t$ depend on $z$ (or $z'$), the Jacobian matrix $\partial x^j / \partial (x')^i$ also cannot depend on $z$. It therefore must have the form

$$\frac{\partial x^j}{\partial (x')^i} = \begin{pmatrix} \frac{\partial x}{\partial x'} & \frac{\partial y}{\partial x'} & \frac{\partial z}{\partial x'} \\ \frac{\partial x}{\partial y'} & \frac{\partial y}{\partial y'} & \frac{\partial z}{\partial y'} \\ 0 & 0 & \sigma \end{pmatrix}, \tag{10}$$

where $\sigma$ is a time-dependent positive constant. Now consider the scalar $m_z$, the $z$ component of $\mathbf{m}$. This is given by $(\alpha_t)_i e_z^i$, where $\mathbf{e}_z$ is the vector parallel to the $z$ direction. Taking the inverse of (10), we find the contravariant $\mathbf{e}_z$ transforms under $\phi_t$ by a simple scale factor of $1/\sigma$. Hence we have the pullback

$$\phi_t^* (m_z(t)) = \phi_t^* \left( (\alpha_t)_i e_z^i \right) = (\phi_t^* \alpha_t)_i \, \phi_t^* (e_z)^i$$
$$= \lambda_t \sigma (\alpha_0)_i e_z^i = \lambda_t \sigma m_z(0). \tag{11}$$

But both $\sigma$ and $\lambda_t$ are positive, hence the sign of $m_z$ is preserved under the diffeomorphism. Converting back to our 2D vector $\mathbf{v}$, we have the following characterization of the structural stability of Skyrmions, or similar twisted solitons with non-zero twisting energy density. The region $S_t$ is deformed over time. Within each region there are a set of (generically) closed curves $\Gamma$ where $v_z = 0$. Over time these closed curves may move within the region $S_t$, but they do not merge, nor are any additional curves created. We include in the set $\Gamma$ the boundary of the Skyrmion, $\partial S_t$, so that one may give a picture of the Skyrmion as a set of closed curves in the plane. This set of curves defines several regions, with $v_z$ always the same sign within each region, as shown in Fig. 2.

This result directly separates a Skyrmionium ($2\pi$ twisted Skyrmion) from the vacuum state as any smooth pathway from one to the other must pass through a configuration where twisting energy density (or contact condition) vanishes at some point.

## 3 Conclusion and discussion

Our theorem shows that a whole class of previously thought non-topological spin textures, such as Skyrmionium, are protected by non-vanishing DM interactions and characterized by a new $\mathbb{N}$ topological invariant. Stability from this topological protection mechanism makes highly-twisted Skyrmions great candidates for building racetrack memories [49] as they will not decay when transported. Moreover, carrying zero net Skyrmion charge, motion of a highly-twisted Skyrmion overcomes the obstacle of transverse deflection due to the Skyrmion Hall effect [50].

Besides ferromagnetic materials, our results equally apply to condensed matter systems exhibiting quantum Hall ferromagnetism. For traditional quantum Hall systems (2D electron gas), there is no spin-orbit coupling term, but impurities, in principle, can introduce such a term acting as a DM interaction [51]. Quantum anomalous Hall ferromagnetic states, such as those found in moiré systems [52–59], might provide a more relevant platform since spin-orbit coupling phenomena form an integral part of the physics of crystals. Furthermore, in the lowest Landau-level, spin degrees of freedom are intertwined with charge degrees of freedom. Skyrmion charge densities from spin configurations(4) are proportional to electric charge densities [16, 18]. Thus, highly-twisted Skrymions, such as Skyrmioniums, in quantum Hall ferromagnetic states can give rise to stable charge neutral excitations. Under a smooth deformation, they can further evolve into neutral excitations with non-zero higher electric moments depending on their shape. Moreover, there are theoretical possibilities of Skyrmions carrying fractionalized degrees of freedom, such as electric charges or Majorana zero modes [17, 60], for potential applications in quantum information processing. Our mechanism could thus provide an extra layer of topological protection for quantum information stored and transported by highly-twisted Skyrmions.

## Acknowledgements

We thank Randall Kamien and Jian Kang for useful discussions.

**Funding information**      This work is in part supported by grant EP/S020527/1 from EPSRC (YH)
.

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
