# Peer review of "Stability of highly-twisted Skyrmions from contact topology"

_SciPost Physics_

## Round 1 · Referee Report · Anonymous (Referee 1) · 2021-7-1

Strengths

(1) The authors make an attempt to use modern mathematical methods to establish the stability of static solitons in models of chiral ferromagnets. These contact geometry/topology methods should allow to make global statements about possible structures and the topological and energetic stability of such vortex-like structures, induced by chiral Dzyaloshinksii-Moriya exchange interactions.

Weaknesses

(1) The arguments to establish stability of these vortices appears to be inconsistent and deficient.

(2) The level of detail as regards the exposition of the mathemetical aspects is inadequate. The authors should clearly state, in which form they apply results from Refs.[45][48] citing the appropriate lemmas and examples and the limitations or restricting factors implied.

The used concepts are not familiar to average theoretical or computational physicists who would like to understand the message of the paper in clear and simple terms.

(3) The specific line of arguments relies on a restricted model for magnetic films with chiral couplings, which cannot exist in ordinary materials.

(3) The authors do not cite crucial relevant literature, in particular recent works from mathematicians who have addressed the stability of radial the standard chiral skyrmions with recent other methods. In particular rigorous results by C. Melcher and co-workers.

(4) The paper is not well constructed. First, in introduction the authors indulge themselves in exposing an analogy between chiral helimagnetic systems with topological solitons having finite and finite excitation energy and the bandstructure of topological insulators and their gaps which is neither obvious nor further substantiated.
In the conclusions, a long section appears about possible other systems where these so-called highly-twisted skyrmions might play a role, which is rather vague and uncertain. Both parts have little relevance for the contents or aims of the main part.

Report

The paper reports an attempt to address a long-standing issue regarding the stability of vortex-like highly-twisted solitons in the classical model for chiral helimagnetic systems. This issue starts with Ref. [3] and the few other earlier papers by Bogdanov et al., which the authors fail to cite. In Ref.[3], Bogdanov et al. present a rather detailed numerical analysis of the continuum model which more or less describes the subject addressed in present manuscript and show data on vortex-like magnetization configurations in 2D with k times PI rotations of the polar angle. In the meantime, the stability of the singly-twisted or PI-skyrmion against small deformations has been established by rigorous methods, cf. Xinye Li, Christof Melcher, Stability of axisymmetric chiral skyrmions, J.Functional Analysis 275, 2817 (2018), but little beyond results of Ref.[3] is known regarding the k PI-vortices.

The authors use a different approach by rather advanced mathematical methods. They claim that their results establish stability of such vortex-like static solitons in a very broad class of further structures, too. The model used is claimed to describe thin ferromagnetic films with chiral exchange and a connection to experiments on such films is made, in particular observations of some rather large concentric skyrmion-bubbles in bubbles in ultrathin magnetic films, Ref. [7].

From the manuscript, however, it is difficult to establish the extent and the validity of the claimed stability results. I find that either their generality is overstated or they are inconsistent with the expectations formed from the numerical studies. The model itself is used in a restricted version that is unphysical, and the line of argument leading to stability claims appears deficient

(i) The authors seem to require a specfic form of the Lifshitz-invariant to establish the DMI-energy as a contact-structure.
Therefore, they choose to consider only the term 'm rot m' or its
2D-analogue. Thus, refering to Eq.(2), they state that their results exlusively apply only to case d_n == 0 and d_b -neq 0. But by symmetry, the standard case of chiral magnetic films has opposite limit, i.e. d_n -neq 0, see the list of possible DMI-terms in Bogdanov, Yablonskii, JETP 1989 depending on point-group symmetries. Generically, all possible invariants should be present in a low-symmetry situation of a thin film. But, the case d_n -neq 0 is impossible. There cannot exist an ordinary magnetic film in 2D with broken inversion symmetry which could realize the case, discussed in this paper. As an curious exception, the theoretical prediction of a very specific certain surface-magnetism may be consulted, Nogueira et al. PRB 98, 0600401 (2018) which might be tuned to this limit. Anyhow, applicability of the results in the manuscript for a magnetic thin film is questionable.

(ii) Further, results seem to rely on an artificial extension of the 2D-model of a film to 3D with homogeneous magnetization structure in the virtual 3d-direction, in order to apply the Gray theorem. But, to this end is appears also necessary to restrict the maps to a disk-like domain M with a fixed value of magnetization vector on its boundary. If this is so, and the text seems a bit unclear with respect to treatment of boundary conditions, then the results seem to pertain only to vortex-like twisted structures in a confined geometry and with a physically less interesting case of a fixed state (magnetization fixed perpendicular to film-plane) at the circular-like boundary. In particular, from these results no statement about the stability of the skyrmion bubbles of Ref. [7] appears possible.

(iii) The claimed generality of the stability of these highly-twisted structures also appears inconsistent with the results of the numerical analysis in Ref.[3] for the same model and the various k PI-vortices or target domains analysed. In this work, it has been shown that such structures can become unstable, e.g. against elliptical strip-out or radial blow-up for certain levels of anisotropy and external field. Physically, these effects are obvious and there is little doubt regarding the results of Ref.[3]. However, in the manuscript such limitations of the stability results are not discussed - and considering the off-hand inclusion of the aE_f-term in model definition, Eq.(1), one may receive the impression that the authors claim that for the whole class of such materials the highly-twisted skyrmions (once injected into a finite region of a film) should remain stable. But, it cannot be true that for these models a highly-twisted skyrmion-vortex would always remain stable and smooth object (even on on a disk-like region) .
On a second reading, it then appears that the authors require that the energetics of the model is such that the chiral part of the energy does not change or remain negative in any case. This appears impossible for the model (1) in full generality, even if one excludes the case of strip-out or blow-up of a solution. Then, one is left to wonder whether the claimed stability of (these metastable excitations) can be real at all, as they might just relax towards an untwisted state, e.g. in the physically really interesting case, when an anisotropy or field favours the collinear state, or into the extended spiral ground-state, if anisotropies are weaker. In both cases, the real physical situation seems to violate the stated requirements for the applicability of the Gray theorem, or the twisted configuration may distort in manner such that it leaves the disk-like region M altogether.

(iii) The identification of contact geometry as an approach for chiral helimagnets is noteworthy - and it may be much more fruitful for 3D bulk (cubic) helimagnets. Unfortunately, the authors do not present any results for such structures and their global topology which might be in reach with this approach.

The presented results and stability claims may still yield some interesting insights. However, the authors should clearly discuss better in which form and how they can apply these concepts. Gray's theorem seems not to imply stability of localized structures for extended spaces R^2 and particularly R^3, i.e. bulk crystals. Thus, the authors should clearly discuss limitations of their approach and make more precise statements regarding the physical situation they want to describe.

Requested changes

The authors should re-consider the validity and limitations of their theorem and the putative results for the physics and stability of such vortex-like structures.

The exposition of the mathematical tools should be vastly improved and clearly connected with the physical situation they address. The authors should keep in mind average physicists as readers:

(i) Without sorting out, how a contact-structure can be identified for the general case of DMIs, as in Eq.(2) with d_n -neq 0, I do not believe the paper should be published.

(ii) The issue of boundary-conditions, radial stability and strip-out in a disk-like domain with fixed boundary should be discussed. Possibly, is is necessary to use more advanced results from contact topology to address the stability of these vortex-like objects in unbounded regions - or other approaches.

(iii) Text and references should be extended to better consider the present state-of-the art, in particular regarding the contributions from mathematics already existing.

---

## Round 1 · Referee Report · Anonymous (Referee 2) · 2021-7-5

Weaknesses

In the current MS, authors use some advanced mathematical methods (contact topology) in order to study the stability of chiral skyrmions. The average theoretical physicist is not familiar with these methods, therefore this MS is hard to read. It would be better to introduce some details on analytical calculations in appendixes or supplemental materials (SciPost does not have any limitation on the length of the paper).

Report

  1. Why you do not consider skyrmions of Neel type? Skyrmions of Neel type are also chiral structures [I], i.e. the sign of the DMI constant defines how in-plane magnetization of a skyrmion is oriented (inward/outward of the skyrmion center). Such skyrmions typically appear in thin films, which is the case of a current MS.

  2. If authors want to consider the DMI in form $\vec{m}\cdot\left[\nabla \times \vec{m}\right]$, which is typical for bulk systems, authors should take into account that order parameter $\vec{m}$ can be a function of third spetial coordinate, i.e. $\vec{m} = \vec{m}(x,y,z)$. In this case one can observe the formation of spherulite structures [II].

  3. Authors should mention the limitations of the considered theory (geometry, size, material parameters....).

  4. Authors claim that their theory is applicable for a wide range of materials (ferromagnets, liquid crystals, ...). In this case it would be nice to add some discussion about antiferromagnets. For instance, how the homogeneous part of the Dzyaloshinskii-Moriya energy $e_\text{dm} ^\text{hom} = \vec{d}\cdot\left[\vec{m}\times\vec{n}\right]$ in AFM system affects your results?

  5. Are skyrmion configurations presented in Figs. 1 and 3 obtained by means of numerical (micromagnetic) simulations or energy minimization? Are deformed skyrmions on Figs. 1(b) and 1(d) stable structures?

  6. In MS there are some typos, e.g. "... Dyloshinskii ..." instead of "... Dzyaloshinskii ..." in abstract.

[I] S. Rohart, A. Thiaville, PRB $\bf{88}$, 184422 (2013) [II] A. O. Leonov, I. E. Dragunov, U. K. Roßler, and A. N. Bogdanov, PRE $\bf{90}$, 042502 (2014)

Requested changes

  1. Add some details on the analytical calculations in appendixes/supplemental materials.

  2. Add details on the theory limitations.

  3. If authors want to consider thin films, so it is necessary to consider skyrmions of Neel type. And for the case of Bloch DMI ($\vec{m}\cdot\left[\nabla \times \vec{m}\right]$) it is convenient to consider 3D structures.

---

## Editorial Decision

awaiting_resubmission